# 400nm ultra-broadband gratings for near-single-cycle 100 Petawatt lasers

Yuxing Han [1,2,3], Zhaoyang Li [4,5] ✉, Yibin Zhang[1,2,3], Fanyu Kong[1,3], Hongchao Cao[1,3], Yunxia Jin [1,3,6] ✉, Yuxin Leng[4], Ruxin Li [4,5] & Jianda Shao [1,3,6,7] ✉

Compressing high-energy laser pulses to a single-cycle and realizing the "$\lambda^3$ laser concept", where $\lambda$ is the wavelength of the laser, will break the current limitation of super-scale projects and contribute to the future 100-petawatt and even Exawatt lasers. Here, we have realized ultra-broadband gold gratings, core optics in the chirped pulse amplification, in the 750–1150 nm spectral range with a > 90% −1 order diffraction efficiency for near single-cycle pulse stretching and compression. The grating is also compatible with azimuthal angles from −15° to 15°, making it possible to design a three-dimensional compressor. In developing and manufacturing processes, a crucial grating profile with large base width and sharp ridge is carefully optimized and controlled to dramatically broaden the high diffraction efficiency bandwidth from the current 100–200 nm to over 400 nm. This work has removed a key obstacle to achieving the near single-cycle 100-PW lasers in the future.

In the past three decades, the race to achieve petawatt (PW) high peak power based on the techniques of chirped pulse amplification (CPA)[1] and optical parametric CPA (OPCPA)[2] has been underway, and two 10-PW lasers have already been demonstrated in Europe and China successfully. Consequently, the next decade will see a dramatic increase in global development work for the delivery of 100-PW lasers[3]. With focused intensities over $10^{23}$ W cm$^{-2}$, these facilities will create new opportunities in secondary source generation[4], particle (electron, proton, ion, etc.) acceleration[5–10], and attosecond science[11]; more importantly, they will present an opportunity to study strong-field quantum electrodynamics (SF-QED)[12] experimentally.

In the journey to a 100-PW or even Exawatt high-peak-power laser, a further reduction in the pulse duration of a single-beamline femtosecond-PW (fs-PW) laser[13,14] is considered an easy and feasible technical route while compared with the spatiotemporally coherent combination of multi-beamline current fs-PW lasers[15]. Recently, a simultaneous chirped beam and chirped pulse amplification method was also proposed to dramatically reduce the pulse duration in a high-energy Nd:Mixed-glass laser for an Exawatt-class output[16]. The "$\lambda^3$" laser, defined by G. Mourou et al.[17], has a single-cycle pulse and a single-wavelength-size focal spot, which is the highest intensity that can be produced by an energy-limited facility. However, up to now, this shortest pulse duration (single-cycle) has not been realized in any PW-class laser, and the key limitation is the narrow spectral bandwidth of laser-amplification media and pulse-compression gratings.

In a Ti: sapphire CPA-PW laser system, the bandwidth of the gain spectrum is around 50–80 nm centered at 800 nm[18–20], which accordingly supports approximately 25–30 fs Fourier-transform-limited pulses. However, because of the limitations in the crystal size and the parasitic lasing[21,22], it is almost impossible to directly enhance the peak power to the 100-PW level by only increasing the pulse energy up to 2500–3000 J. Further, in a BBO/LBO-based OPCPA laser system, the gain bandwidth could exceed 200 nm[23] centered at 800 nm. However, the small crystal size cannot support the high energy

[1]Laboratory of Thin Film Optics, Shanghai Institute of Optics and Fine Mechanics, Chinese Academy of Sciences, Shanghai 201800, China. [2]Center of Laboratory of Materials Science and Optoelectronics Engineering, University of Chinese Academy of Sciences, Beijing 100049, China. [3]Key Laboratory of Materials for High Power Laser, Chinese Academy of Sciences, Shanghai 201800, China. [4]State Key Laboratory of High Field Laser Physics, Shanghai Institute of Optics and Fine Mechanics, Chinese Academy of Sciences, Shanghai 201800, China. [5]Zhangjiang Laboratory, Shanghai 201210, China. [6]CAS Center for Excellence in Ultra-Intense Laser Science, Chinese Academy of Sciences, Shanghai 201800, China. [7]Hangzhou Institute for Advanced Study, University of Chinese Academy of Sciences, Hangzhou 310024, China. ✉e-mail: lizy@zjlab.ac.cn; yxjin@siom.ac.cn; jdshao@siom.ac.cn

amplification[24]. Several institutions in Europe, the US, and China have proposed that tens-PW ultra-intense lasers could be generated by the large-aperture DKDP-based OPCPA with around 200 nm bandwidth centered at 910 or 925 nm[25–27]. To realize the 100-PW laser with the current energy level, several schemes are proposed to reduce the compressed pulse by broadening the spectral bandwidth of OPCPA up to around 400 nm, such as achromatic phase-matching optical parametric amplification (OPA)/OPCPA[28], OPCPA with DKDP at low deuteration[29,30], cascaded/parallel-channel OPCPA[31–36], wide-angle non-collinear OPCPA (WNOPCPA)[13,14], and so on. Another key obstacle is the large-aperture broadband grating for pulse compression. The current advanced gold grating has a pulse-duration-insensitive laser-induced damage threshold (LIDT) when below 20 fs[37,38] and the meter-sized large aperture; however, its narrow bandwidth cannot support the single-cycle pulse compression.

Based on the above system architecture design, standard line densities, for example, 1480 and 1740 lines/mm gold gratings, produced by LLNL[39], Horiba[40], PGL[41], SUDA[42], and SIOM[43,44], were available in high-energy 800 nm-centered laser systems with around 100 nm bandwidths. Adjusting the groove depth and profile, the −1 order high-diffraction bandwidth could be improved to about 200 nm centered at 910 nm or 925 nm. However, no evidence shows that the ultra-broadband grating design and fabrication could support a 400 nm bandwidth, which is desperately required by the concept of the single-cycle 100-PW laser. Some engineering planning requires improper line density and angular spectrum of the gold grating. Accordingly, the available range of grating parameters needs to be specified. Moreover, there is little work examining the evolution behavior of groove profiles for critical process parameters[45,46].

Until now, most pulse compressors operate in the non-azimuthal angle configuration (two-dimensional compressor), except for a few projects[47]. With the increase in spectral bandwidth demand, the application of non-azimuthal angle configuration for gold gratings or multilayer dielectric gratings[48] hits a bottleneck in avoiding the spatial overlap of laser beams. The pulse compressor with an azimuthal angle (three-dimensional compressor) will become an added trend for a further upgraded version of active laser systems[49].

Here, we present the 400 nm ultra-broadband gold grating compatible with large azimuthal angles developed for near-single-cycle pulse stretching and compression. Tolerance analysis is performed to redefine the available grating parameters in the engineering layout. The proposed groove design requires a large base width and a sharp ridge that can be obtained by adjusting the strength of the developer. All process parameters are described in detail. The grating exhibits a high −1 order diffraction efficiency greater than 90%, covering >400 bandwidth from 750 nm to >1150 nm, which can facilitate the development of next-generation high-peak-power laser systems for SF-QED[50].

## Results

### Design and analysis

The grating compressor is a core setup in a CPA or OPCPA PW laser, where the amplified high-energy long chirped pulse is temporally compressed to an ultra-high peak-power short pulse. Fig. 1 presents a schematic of the grating diffraction spectral bandwidth that determines the shortest duration and the highest peak power of the compressed pulse.

Holographically produced gold grating gives rise to a characteristic groove profile approximated by

$$h(x) = h_{max} \cdot \max\left\{0, 1 - \left[\frac{\cos^2(\pi x/\Lambda)}{\sin^2(\pi f/2)}\right]^\sigma\right\} \quad (1)$$

where $h_{max}$ denotes the groove depth, $\Lambda$ indicates the grating period, $f$ symbolizes the duty cycle in the base (the base width is the product of

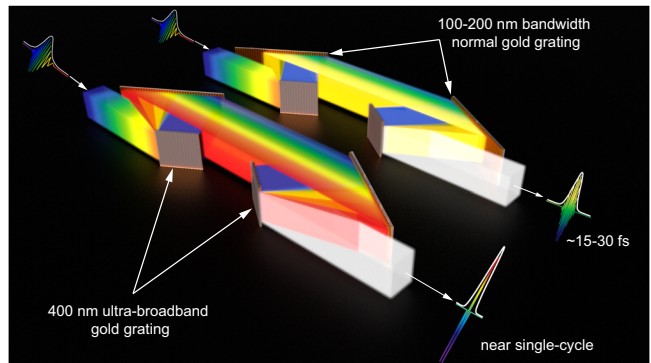

**Fig. 1 | Schematic of ultra-broadband grating compression effect.** Grating diffraction bandwidth determines the compressed pulse duration and intensity of PW lasers. At the same 100-PW system architecture deployed with the normal gold grating, a 400 nm ultra-broadband grating compressor enables a near-single-cycle Exawatt-class (>200 PW) pulse compression.

the duty cycle and the period), $\sigma$ represents the shape factor, and $x$ corresponds to the lateral dimension. In the ultra-broadband gold grating design process, the abovementioned adjustable parameters are liberalized to search for the optimum groove structure. The diffraction efficiency is computed for a given spectral domain using a rigorous coupled-wave analysis code for each groove profile given by Eq. (1).

Consequently, the final selected design is illustrated in Fig. 2a. The ultra-broadband gold grating is intended to have 1443 lines/mm, for which the incident angle is 50°, and the shape corresponds to $f = 0.634$, $h_{max} = 225$ nm, and $\sigma = 1.91$. The ultra-broadband diffraction efficiency spectrum presents the double-acromion state, which is a typical result for deviations from the Littrow angle. The calculated result indicates that the diffraction efficiency can exceed 90% in the 724−1198 nm wavelength range. When the designed grating was applied in a recent WNOPCPA sub-Exawatt laser design for ultra-broadband high-energy pulse stretching and compression (see Supplementary Note 1), the diffraction efficiency curve (Fig. 2b, pink curve) completely covers the ultra-broadband gain spectrum (Fig. 2b, blue curve). In this case, Fig. 2c indicates that the Fourier-transform-limited pulse is 6 fs (FWHM), only containing around two cycles of the carrier frequency. Moreover, if considering a sufficiently broadband input pulse adaptive with the full diffraction spectrum, the designed grating can support a 4-fs (FWHM) Fourier-transform-limited pulse (Fig. 2d), very near a single cycle of -3.3 fs.

As depicted in Fig. 3a, in the non-azimuthal angle architecture, a narrow angular spectrum (56−61°) is insufficient for engineering. In order to expand the incident angle spectrum, the azimuthal angle is introduced into the optimization. Fig. 3b shows that ultra-broadband gold grating is compatible with a broad azimuthal angle range in (0, ±15°), where the depolarization effect cannot deteriorate the bandwidth and efficiency. Consequently, the 400 nm bandwidth over 90% requirements can achieve an incident angle range of 45−61° in the azimuthal angle architecture.

In practice, it is desirable to have a design with relatively little sensitivity to minor variations in design parameters. Fig. 4 depicts how the line density, duty cycle, shape factor, and depth affect the grating efficiency in the above design. In Fig. 4a, the broadband diffraction efficiency exhibits a weak dependence on the line density. In the extensive line density range, from 1400 to 1500 lines/mm, the diffraction efficiency is higher than 90% in the 750−1150 nm. Importantly, laser designers should note that while diffraction efficiency is relatively insensitive to line density, it is best to limit it to a specific range considering the issue of light blocking. Fig. 4b displays two striking features. First, the high-efficiency bandwidths cover 400 nm when the

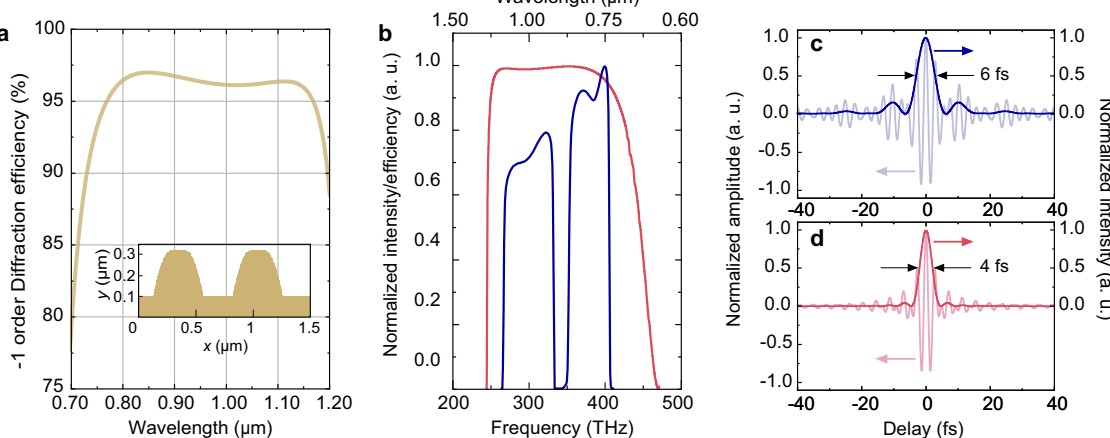

**Fig. 2 | Ultra-broadband gold grating and a WNOPCPA sub-Exawatt laser design. a** Calculated −1 order diffraction efficiency as a function of wavelength for the simulated grating profile shown in the inset picture, used at an incident angle of 50° with TM polarization without an azimuthal angle. The shape corresponds to $\Lambda = 693$nm, $f = 0.634$, $h_{max} = 225$nm, and $\sigma = 1.91$. **b** Diffraction efficiency curve of grating [pink curve] can support the gain spectrum [blue curve] and (**c**) accordingly 6 fs Fourier-transform-limited pulse of a WNOPCPA 100 PW design. **d** Fourier-transform-limited pulse of grating diffraction efficiency curve is 4 fs (near single cycle). In (**c**, **d**), both pulse intensities and carrier frequency amplitudes are presented.

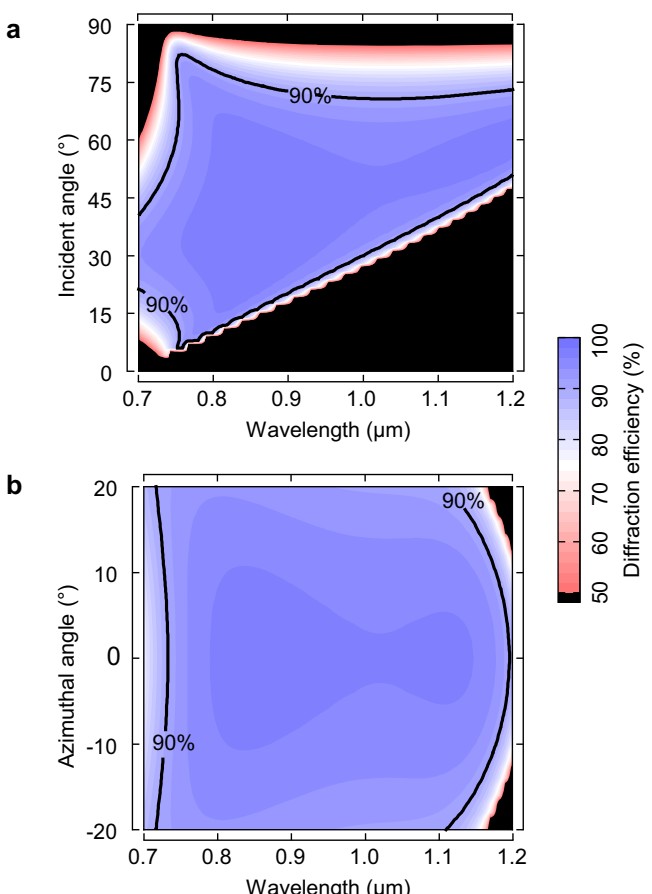

**Fig. 3 | Angle tolerance.** Calculated −1 order diffraction efficiency as a function of (**a**) incident angle without an azimuthal angle and (**b**) azimuthal angle at 50° incident angle for the simulated grating profile shown in Fig. 2.

duty cycle is greater than 0.6. Second, the efficiencies at the long wavelengths (>1 μm) decrease as the duty cycle increases. Fig. 4c emphasizes that the shape factor should be controlled between 1.3 and 2. Fig. 4d suggests a depth tolerance range between 183 and 235 nm.

The wide range of parameter choices can satisfy the primary design objective of high diffraction efficiency.

## Grating profile evolution

The diversity of solutions presents an opportunity for imposing additional requirements on the fabrication. The process of ultra-broadband gold grating in SIOM was explored (see "Method" section).

Many trials were performed to explore the precise connection between the controllable factors and the groove profile. The results were divided into two rounds based on the strength of the developer. In Round I and II, photoresist gratings were developed with 4 wt.‰ NaOH solution for 50–90 s and 6 wt.‰ NaOH solution for 5–8 s, respectively.

Fig. 5 summarizes the mapping relationship between the efficiency spectrum, grating profiles, and developing time. Several features are of significance. First, as shown in Fig. 5a, the efficiency traces of S2 and S3 are high and overlap in the long-wavelength range (950–1150 nm). The groove profiles of S2 and S3 have similar sharp ridges with a shape factor close to 2.2 with a depth of -214 nm (Fig. 5c, d, and f-shape factor). However, compared with S3, the efficiencies of S2 in the short-wavelength band (750–950 nm) are significantly improved, which is the contribution of the larger base width (Fig. 5d and f-duty cycle). Second, S4 presents the broadest bandwidth, interpreted as the closest base width and groove depth values (Fig. 5f) to the theoretical design value (Fig. 2a). Conversely, the efficiency level of S4 assumes a downtrend from short to long wavelengths owing to the flattop ridge (Fig. 5e). Third, S1 is treated for the shortest developing duration, leading to partially etched grooves. S1 performs well at short wavelengths owing to the large base width with a duty cycle up to 1. However, the diffraction efficiency gradually drops out of the 90% baseline in the long-wavelength range, resulting from the flat ridge with a shape factor up to 3.26 and a depth of only 205 nm (Fig. 5b and f-depth). Round I confirms that it is feasible to increase the base width to broaden the bandwidth of the high diffraction efficiency.

Nevertheless, the groove profile with a shape factor above two is consistently obtained in the 4 wt.‰ NaOH solution developing scheme, which is somewhat flat for the design profile. Accordingly, it was decided to use 6 wt.‰ NaOH solution in Round II. As presented in Fig. 6a, with the increase in the development duration, S6, S7, and S8 make dramatic advances in efficiency and bandwidth compared with S5. AFM and FIB-SEM results reveal that the depths of the four samples

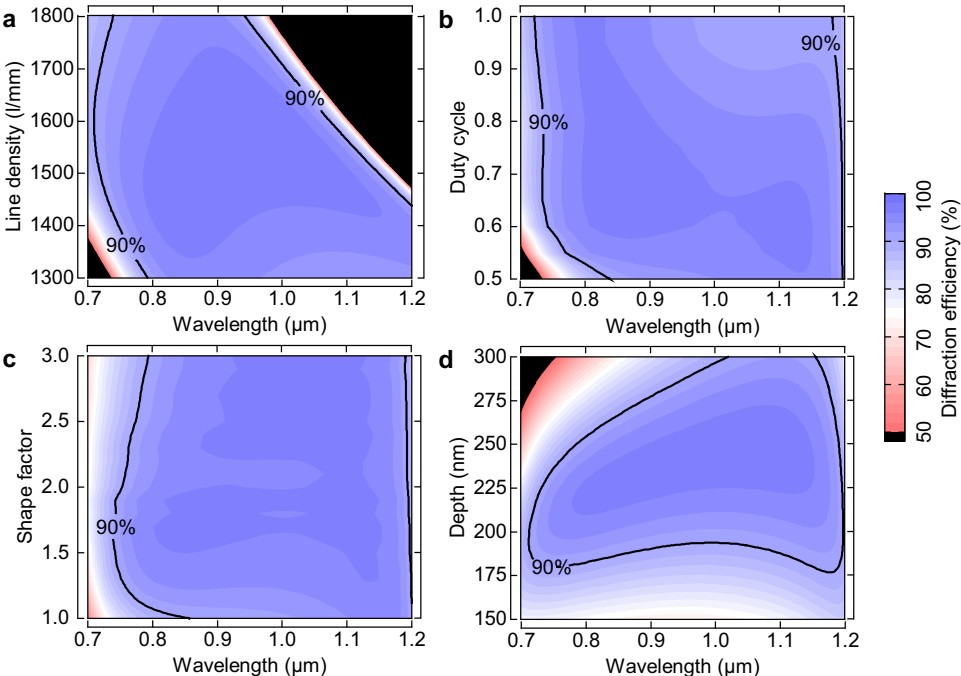

**Fig. 4 | Structure tolerance.** Dependence of the diffraction spectrum on (**a**) line density, (**b**) duty cycle, (**c**) shape factor, and (**d**) depth of the ultra-broadband grating. The wavelength range of light is 0.7–1.2 μm at 50° with TM polarization.

are all approximately 222 nm. The duty cycle of S6, S7, and S8 are approximately 0.6 (Fig. 6c–e, and f-duty cycle), which is close to the design value. However, the shape factor significantly differs from 2.36 to 1.29 (Fig. 6f-shape factor). Compared with S7, the high diffraction efficiencies in the long-wavelength band of S6 and S8 are attributable to deeper groove depths (Fig. 6a and f-depth). The flat ridge with a shape factor beyond two results in low diffraction efficiencies of S5 and S6 in the 750–810 nm range (Fig. 6a and f- shape factor). Consequently, the groove profile reveals that constructing a small sharp ridge could maintain high diffraction efficiency in the long-wavelength band while increasing the diffraction efficiency in the short-wavelength band.

## Discussion

According to the evolution law of the groove profile mentioned above, the duty cycle and shape factor are the most sensitive parameters for improving the diffraction efficiency at short wavelengths in manufacturing ultra-broadband gold grating. Fig. 7 shows the dependence among the duty cycle, shape factor, and diffraction efficiency at 750 nm. The profile parameter distribution of samples illustrates that combining a large base width and a small sharp ridge is required for high diffraction efficiency at 750 nm.

Based on the trend of profile parameters in S1–S8, S9 is developed with 5.5 wt.‰ NaOH solution for 12 s, resulting in a groove profile with a 0.68 duty cycle, 200 nm depth, and 1.83 shape factor (Fig. 7 S9; Fig. 8b 1443 lines/mm). Moreover, 1443 lines/mm ultra-broadband gold gratings are demonstrated over 750–1150 nm at 50° with TM polarization (Fig. 8a blue line). In addition, a scanning diffraction efficiency map was measured at the center wavelength of 950 nm (see "Method" section). The area with higher than 90% diffraction efficiency accounts for 95.5% of the total measured area, and the average efficiency is 92.7% with a standard deviation of 0.99% (Fig. 8c left map).

Furthermore, the expanding diffraction light in space needs to preserve a specific misalignment with incident light in the architecture of the grating compressor without an azimuth angle. Accordingly, another ultra-broadband gold grating was developed to encompass 1527 lines/mm, for which the incident angle is 62°, and the profile corresponds to $f = 0.765$, $h_{max} = 216nm$, and $\sigma = 1.36$ (Fig. 8b 1527 lines/

mm). Photoresist grating was developed with 6 wt.‰ NaOH solution for 12 s, and other process parameters are the same in the Method section. The measured result indicates that the diffraction efficiency can reach over 90% in the 742–1150 nm wavelength range (Fig. 8a red line). Moreover, the efficiency map is extremely uniform over the central 93% of the grating area, exhibiting an average efficiency of 93.8% with a standard deviation of 0.73% (Fig. 8c right map).

Meanwhile, the 1-on−1 damage test (see "Method" section) was performed to investigate the initial failure state of the ultra-broadband gratings in the front end for the Station of Extreme Light (SEL). The front end provides a laser pulse with the widest spectral width and the shortest pulse duration among all the 100-TW-level OPCPA systems at present[26]. 1443 and 1527 lines/mm ultra-broadband gold gratings exhibit the surface LIDT of 350 mJ cm$^{-2}$ and 272 mJ cm$^{-2}$ (Fig. 8d), respectively. These values are sufficient to meet the LIDT requirements of current compressors. Future work aims to perform an LIDT test on the meter-scale grating with a more broadband laser in the vacuum.

The peak power of a multi-PW-class ultra-intense laser is determined by the bandwidth, aperture, and LIDT of compressor gratings, which is approximately given by

$$P = \frac{E}{\tau} = \frac{S \cdot \cos \theta \cdot F_{\text{surface}}}{\tau} \qquad (2)$$

Here, $P$ denotes the laser peak power, $E$ indicates the pulse energy, $\tau$ symbolizes the pulse duration, $S$ signifies the laser aperture, $\theta$ corresponds to the incident angle, and $F_{\text{surface}}$ indicates the surface LIDT of the gold grating. Considering the different experimental conditions of worldwide research groups, the surface LIDT of the gold grating is limited to 100 mJ cm$^{-2}$. Referring to a current compressor design in a 15 fs-750 J-50 PW laser project: 1400 lines/mm density density and 61° incident angle, the grating aperture is up to 1410 mm (length) × 660 mm (height)[51]. If the above laser project is further upgraded to 15 fs−1500 J−100 PW, the compressor requires gratings with an aperture of 1620 mm (length) × 1070 mm (height).

Since the 400-nm ultra-broadband gold grating can support much shorter pulses of 4–6 fs, 100-PW ultra-intense lasers can be

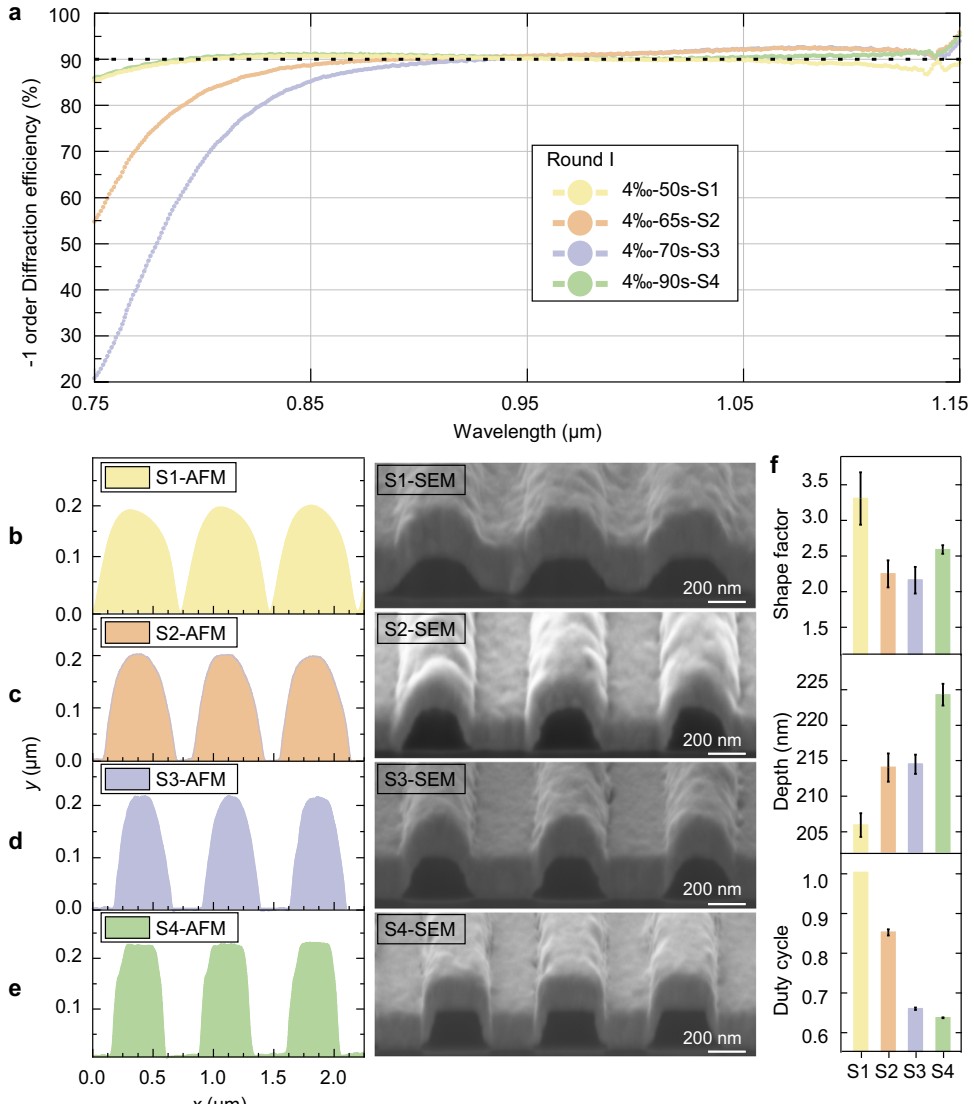

**Fig. 5 | Grating profile evolution with 4 wt.‰ NaOH solution. a** Measured diffraction efficiencies of samples in Round I at 750–1150 nm. **b**–**e** Grating groove profile measured by AFM and FIB-SEM. The curves are labeled according to developer concentration, developing time, and sample number. **f** Structure parameter inversion for S1–S4. The error bar stands for the statistical error.

achieved with much lower energies of 400–600 J, or the required grating aperture decreases from nearly two meters down to around one meter (see Supplementary Note 1), which reduces manufacturing difficulties and saves costs. When loading the pulse energy to the 1-kJ level and installing the meter-scale gold gratings with a 400-nm spectral bandwidth, the peak power will upgrade to the Exawatt-class. Currently, the large-aperture ultra-broadband gold grating fabrication campaign is well underway, and the manufacturing capability can support the 400 nm bandwidth (see Supplementary Note 2). The effects of the grating diffraction efficiency map and wavefront on the spatiotemporal structure of the compressed and focused pulsed beams were simulated (see Supplementary Notes 3 and 4, respectively) without introducing any significant distortion.

In conclusion, the ultra-broadband gold grating in the 750–1150 nm range was developed and demonstrated experimentally. The groove design proposed in this article requires a large base width and sharp ridge. More specifically, 1443 and 1527 lines/mm ultra-broadband gold gratings were developed successfully for compressors, both with and without azimuthal angles, respectively. All process parameters were described in detail. A 4 wt.‰ NaOH

developer can be used to construct a large base width to broaden the bandwidth; however, it was constrained by flattop ridges. Conversely, a 6 wt.‰ NaOH developer shapes sharp ridges to enhance the efficiency of the long-wavelength band. The results and analysis are compatible with an extensive range of line densities. The developed ultra-broadband grating enables high-fidelity pulse stretching and especially pulse compression for realizing near-single-cycle 100-PW lasers, which would benefit many fields ranging from high-field physics to ultra-fast sciences.

## Methods
### Ultra-broadband gold grating fabrication
The gratings were fabricated to the specifications of Fig. 2. Ultra-broadband gold grating was prepared on a 50 mm × 50 mm × 1.5 mm optically flat fused silica substrate. Accordingly, 200–240 nm thick photoresist films were coated to substrates using a spin coater at a speed of 2500 r min⁻¹ for 30 s before baking for 2 min at 100 °C. Following soft-baking of the photoresist films, the blanks were exposed to 325 nm light with 50–60 μW cm⁻² exposure power for 200 s. The latent-image photoresist gratings were developed with a specific

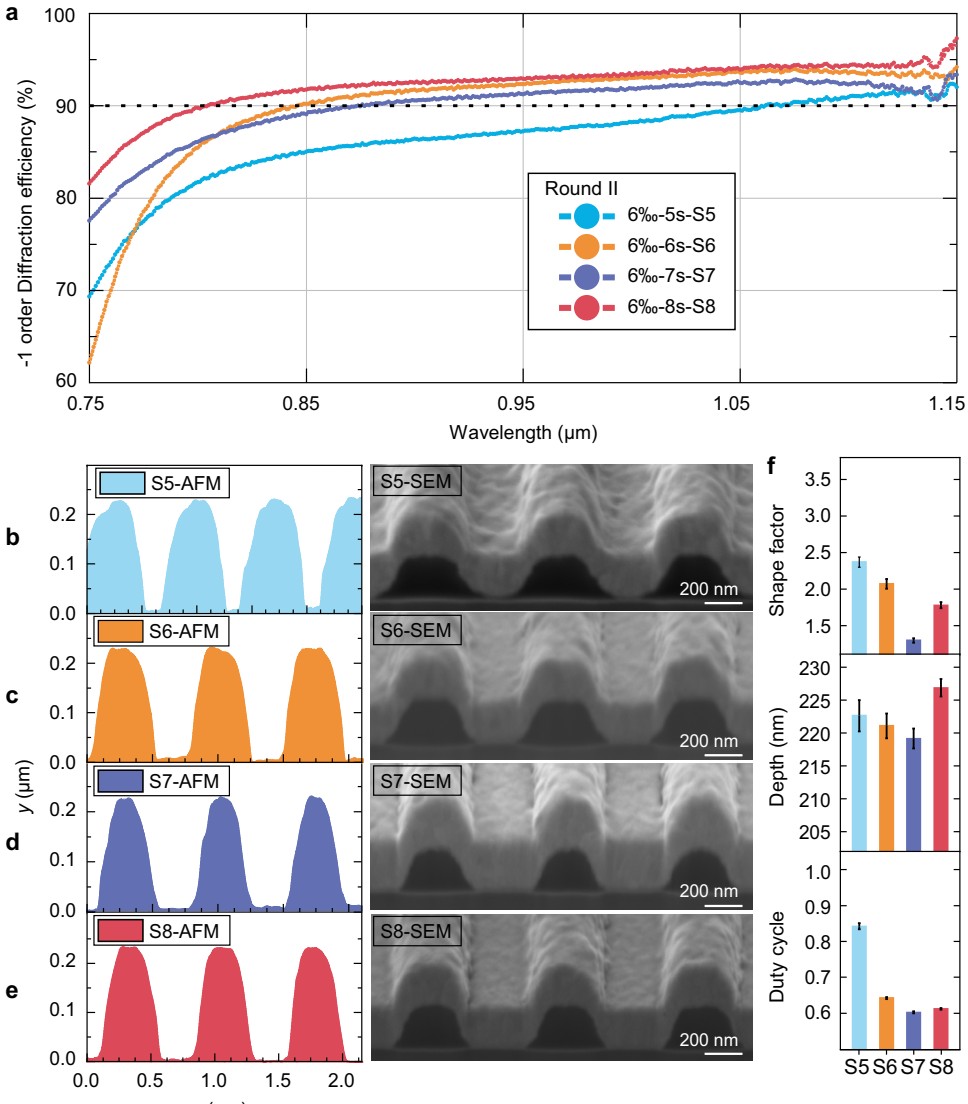

**Fig. 6 | Grating profile evolution with 6 wt.‰ NaOH solution. a** Measured diffraction efficiencies of samples in Round II at 750–1150 nm. **b**–**e** Grating groove profile measured by AFM and FIB-SEM. The curves are labeled according to developer concentration, developing time, and sample number. **f** Structure parameters inversion for S5–S8. The error bar stands for the statistical error.

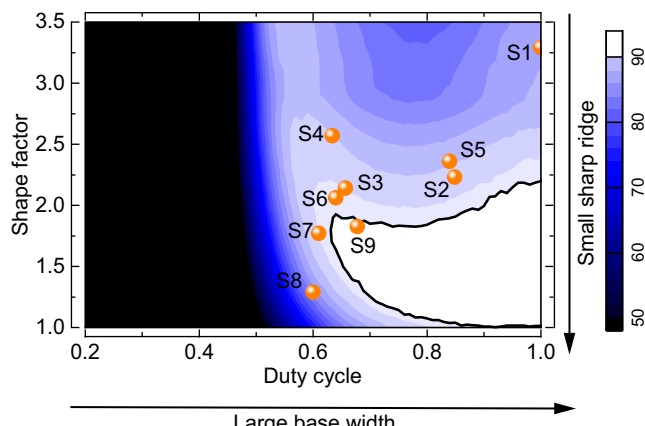

**Fig. 7 | Large base width and small sharp ridge.** Dependence of the diffraction efficiency on the duty cycle and shape factor. The wavelength of light is 0.75 μm at 50° with TM polarization. The orange spheres represent the parameter distribution of samples in Rounds I (S1–S4), II (S5–S9), and the final product (S9).

concentration of NaOH solution. The photoresist gratings were overcoated with a 10 nm Cr adhesion layer and a 150 nm gold coating by magnetron sputtering.

## Broadband diffraction efficiency characterization

The diffraction efficiency measurement setup for the ultra-broadband gratings mainly consisted of a double light-path system with a monochromator as the illumination source and an automatic rotation stage for adjustment of incident and diffraction angles. The high-spectral-resolution diffraction efficiency spectra in the range of 700–1100 nm and 1100–1150 nm were measured by a Si detector (OPHIR PD300R-UV) and a Ge detector (OPHIR PD300R-IR), respectively. The wavelength resolution is 1 nm, and diffraction efficiency has an average relative deviation of less than 1.0% (see more details in Ref. [52]). In a scanning photometry map[53], the grating was measured over an area of 46 mm × 46 mm with a laser beam size of 2 mm and a step size of 2 mm. Each diffraction efficiency map contains 576 points, of which the number of single-point sampling is 5. 1443 and 1527 lines/mm ultra-broadband gold gratings were tested at 50° and 62° in TM polarization, respectively.

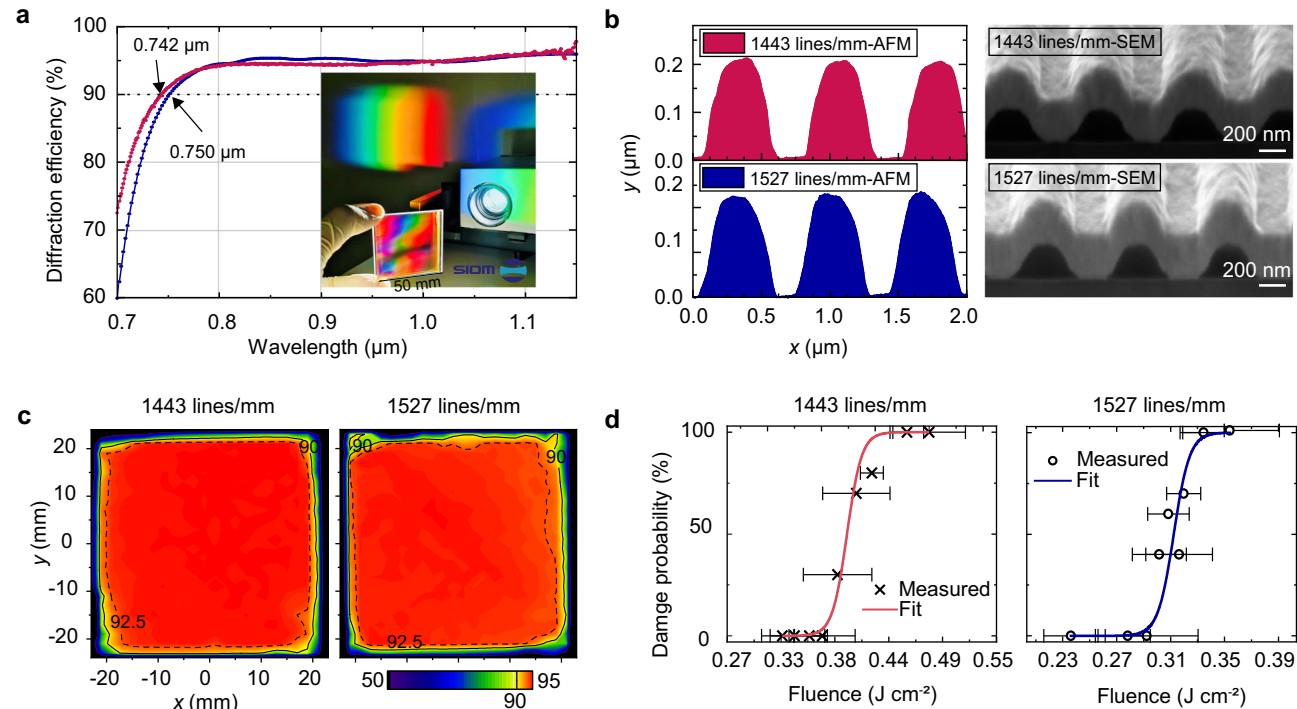

**Fig. 8 | 400 nm ultra-broadband gold grating. a** Measured −1 order diffraction efficiency of 1443 lines/mm at 50° (red line) and 1527 lines/mm (blue line) at 62°, both with TM polarization. An ultra-broadband gold grating is shown in the inset pictures. **b** Measured grating profiles. **c** Scanning photometric diffraction efficiency map at 950 nm. **d** 1-on−1 damage probability as a function of the input fluence. The error bar stands for the statistical error.

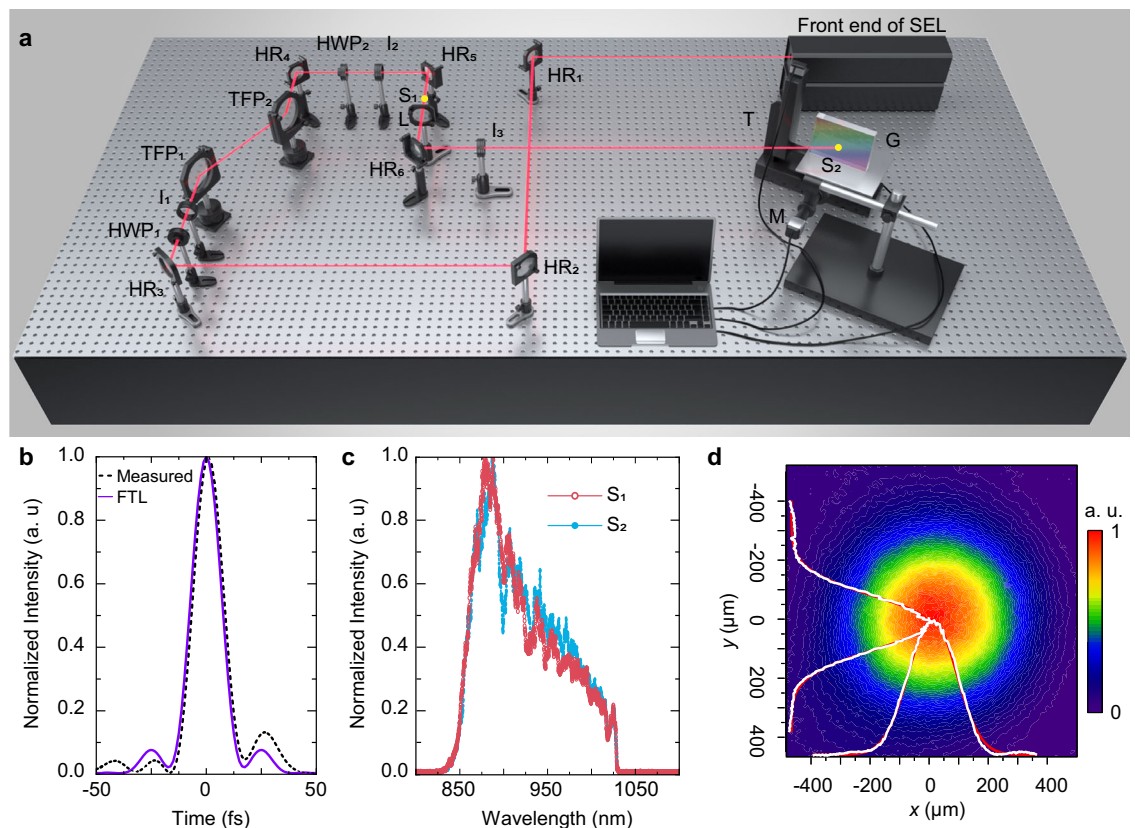

**Fig. 9 | Damage-test table in the front end for the station of extreme light (SEL). a** Layout of the test facility. HR High reflectivity mirror, HWP half-wave plate, TFP thin-film polarizer, L lens, I iris, T translation stage, M microscope, G grating, S site (indicated by two yellow points). The pulse width, spectrum, and beam profile were measured in $S_1$ or $S_2$. **b** Measured and FTL pulse duration of the compressed laser pulse in $S_1$. **c** Measured spectrum in $S_1$ and $S_2$. **d** Beam profile in $S_2$.

## Grating profile measurement

The surface and cross-section morphologies of all gratings are obtained by atomic force microscopy (AFM, Bruker Dimension Icon) with a contact mode in air and a high-resolution focused ion beam scanning electron microscope (FIB-SEM, Carl Zeiss AURIGA CrossBeam).

## Laser-induced damage test

The LIDT tests of the 400-nm ultra-broadband gold gratings were assessed in SIOM. The layout of the damage test was developed from the front end for the SEL, as shown in Fig. 9a. This front end was an LBO-based OPCPA system, capable of generating a 13.4 fs pulse with a bandwidth over 200 nm near the center wavelength of 925 nm at 0.1 Hz.

A pair of 200 nm broadband thin-film polarizers has been self-developed with a 90: 1 extinction ratio. An achromatic half-wave plate (Thorlabs AHWP10M-980) with two polarizers acted as a fluence-variable-ratio attenuator. Another half-wave plate further controlled the laser polarization. For the damage test, the beam was focused with a 1500 mm long focal-length lens. The beam profile on the focal plane was imaged onto a charge-coupled device (CCD, Ophir SP907), and the pulse duration was measured by a Wizzler (Fastlite, USP4). The gratings were mounted on a motorized translation stage (LianYi XZM200H −150) at the focus. During the test, the irradiated sites of gratings were observed using a long working distance and high magnification zoom microscope objective equipped with a CCD camera (XiangQue MetOPT800). A self-programmed procedure controlled the damage test setup.

As shown in Fig. 9b, the compressed pulse width is $15 \pm 1$ fs in $S_1$. Self-phase modulation (SPM) must be considered for the design of high-peak-power damage-test systems. The spectrum (Fig. 9c) in $S_1$ and $S_2$ shows a relatively weak self-phase modulation effect. The effective beam size is ~0.07 mm$^2$ in the beam normal.

According to the ISO-21254 LIDT standard, 1-on-1 damage tests were performed in the air. 1443 and 1527 lines/mm gratings were tested at 50° and 62° in TM polarization, respectively. Twelve sites were tested for each energy fluence. Note that the LIDTs were provided on the surface and determined by extrapolating the damage probability curve to zero probability.

## Data availability

Relevant data supporting the key findings of this study are available within the article and the Supplementary Information file. All raw data generated during the current study are available from the corresponding authors upon request.

## Code availability

The code that supports the plots within this study is available from the corresponding authors upon request.

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

## Acknowledgements

This work is supported by National Key R&D Program of China (Grant Number 2020YFA0714500 to Y.J.), National Natural Science Foundation of China (Grant Numbers 61875212 to H.C. and U1831211 to J.S.), and Shanghai Strategic Emerging Industry Development Special Fund (Grant Number 31011442501217020191D3101001 to Y.J.). The authors thank Dr. Xinliang Wang (the State Key Laboratory of High Field Laser Physics of SIOM) and Dr. Xinyan Liu (Zhangjiang Laboratory) for providing the laser source in damage tests. The authors gratefully acknowledges Dr. Yun Cui (Laboratory of Thin Film Optics of SIOM) for the assistance with FIB-SEM measurements.The scanning photometry map of the diffraction efficiency were acquired at the Precision Optical Manufacturing and Testing Center of SIOM.

## Author contributions

Y.H., Z.L., Y.J., and J.S. conceived the original concept and initiated the work. Y.H. performed the theoretical analysis and conducted the simulations. Y.H. and Y.Z. fabricated the samples. Y.H. characterized the samples. Y.Z., F.K., and H.C. developed the exposure setup. Y.H., Y.Z., Y.J., Z.L., F.K., and H.C. discussed the results. Y.H. wrote the manuscript, and all authors reviewed the manuscript. R.L, Y.L, Y.J and J.S. supervised the project.

## Competing interests

The authors declare no competing interests.
