## [Peer Review File · Nature Communications]

400 nm ultra-broadband gratings for near-single-cycle 100 Petawatt lasersREVIEWER COMMENTS

Reviewer #1 (Remarks to the Author):

The manuscript by Han et al. discusses an extension of the gold diffraction grating design that can achieve a broad spectral bandwidth suitable for the compression of pulses in chirped-pulse amplification (CPA) systems to much shorter pulse durations than possible with today's commercially available gratings. The main idea is centered around the parameter search for a groove profile and optimization of the fabrication method to achieve this profile. An accompanying simulation of pulse compression in a CPA system that uses an optical parametric amplifier as a gain medium (OPCPA) is performed.

While the authors demonstrate grating samples that indeed show efficiency characteristics compatible with few-cycle pulse compression, this work does not meet the standards of publication in a high-impact journal. This is for several reasons:

1. The authors fabricated only a 50-mm grating sample, with an area that is orders of magnitude smaller than what is needed for the ultimate application (for which gratings in requisite sizes were demonstrated in the 1990s).
2. There is no evidence for uniformity of diffraction efficiency.
3. The authors never attempted to compress a broadband laser pulse with this grating and measured its pulse contrast. It is far from obvious that the surface quality of this grating could support compression into such a short pulse without producing a group delay ripple that greatly affects the peak power and could cause excessive spatiotemporal distortion.
4. The authors did not measure the laser damage threshold for their samples, which is critical for the proposed use case.

In conclusion, this is an interesting and encouraging result that offers a hint of the possible direction in broadband grating development for CPA, and this reviewer is of the opinion that it will be of interest to the audience of a more specialized journal.

Reviewer #2 (Remarks to the Author):

The manuscript “400 nm ultra-broadband gratings for near-single-cycle 100 Petawatt lasers” by Yuxing Han et al. provides a new grating design intended for scaling peak powers beyond current 10-PW-level systems by allowing the compression of much larger bandwidths potentially reaching the 100-PW-level. This detailed design and fabrication methods presented here would prove useful to advancing grating technology and benefiting the laser community. I recommend this manuscript for publication in Nature Communications.

Summary:

The authors reference key CPA techniques such as Ti:Sa CPA, BBO OPCPA, DKDP OPCPA, and wide-angle noncollinear OPCPA and some of their limitations. Reducing the pulse duration is identified as one of the more straight-forward ways to significantly boost peak power since current 10 PW lasers are nowhere near a single-cycle pulse duration. Current grating technology can support ~200 nm bandwidths in the near-infrared, but this new design will allow improved broadband diffraction efficiency for reaching single-cycle pulse durations. This article presents a design for 400 nm ultra-broadband gold gratings compatible with large azimuthal angles (ie. out-of-plane compressors) developed for near-single-cycle pulse stretching and compression. The authors identified free parameters that may be adjusted in the grating design and proposed a new design based on an optimized parameter space. In addition to optimization the tolerancing was also considered to ensure that the design was realistic to fabricate with common methods. The grating profile evolution was thoroughly studied to find the most reliable fabrication recipe. A 4 wt.% NaOH developer was found to broaden the base width while a 6 wt.% NaOH developer allowed the sharpening of ridges, both of which contributed to enhancing the efficiency over a broader bandwidth. The grating design was incorporated in a wide-angle NOPCPA system that could produce pulses as short as 6 fs, where the diffraction efficiency of the gratings could ideally compress a bandwidth sufficient for a 4 fs FTL pulse duration. The gratings exhibit a high diffraction efficiency > 90% for >400 nm bandwidth from 750 to 1150 nm, facilitating development of next-generation high-peak-power lasers.

Comments:

- Research is thorough and well-organized with appropriate figures showing the grating design compared to measurements of the fabricated gratings.
- The proposed grating design with optimized base width, groove depth, and shape factor is backed up with quality characterization and experimental implementation in an ultra-broadband OPCPA system.
- The fabrication and characterization methods are described in clear detail, which is very useful to others working to develop improved gratings for ultra-broadband lasers.
- The quality of the manuscript text and figures are high.

SHANGHAI INSTITUTE OF
OPTICS AND FINE MECHANICS
CHINESE ACADEMY OF SCIENCES
Shanghai 201800 CHINA

P. O. Box 800-211 Shanghai
Tel: 0086-21-69918000
Fax:0086-21-69918800
<http://www.siom.cas.cn>

Answers to Reviewer 1

Reviewer #1 (Remarks to the Author):

The manuscript by Han et al. discusses an extension of the gold diffraction grating design that can achieve a broad spectral bandwidth suitable for the compression of pulses in chirped-pulse amplification (CPA) systems to much shorter pulse durations than possible with today's commercially available gratings. The main idea is centered around the parameter search for a groove profile and optimization of the fabrication method to achieve this profile. An accompanying simulation of pulse compression in a CPA system that uses an optical parametric amplifier as a gain medium (OPCPA) is performed.

While the authors demonstrate grating samples that indeed show efficiency characteristics compatible with few-cycle pulse compression, this work does not meet the standards of publication in a high-impact journal. This is for several reasons:

Question 1. The authors fabricated only a 50-mm grating sample, with an area that is orders of magnitude smaller than what is needed for the ultimate application (for which gratings in requisite sizes were demonstrated in the 1990s).

Answer 1: Thanks for the comment and query.

1. Absolutely! As stated by reviewer 1, the steady ascent of Ti:sapphire, OPCPA, and Nd:glass technologies upward in peak power has, with the construction of 1- to 10-PW systems, required the meter-scale gold grating.

In 1996, **940-mm** gold grating¹ developed by LLNL facilitated the first-ever PW laser, Nova. In 2017, **970-mm** gold grating co-produced by THU, USTC, SUDA, and SIOM was installed in XG-PW². In 2019 and 2022, up to **1000-mm** gold gratings from Horiba Jobin Yvon were sold to SULF³ and EFL-NP⁴ with a specification of 10 PW.

Undoubtedly, our ultimate goal is to manufacture meter-scale pulse compression gratings for 100-PW laser facilities. As early as 2017, our group had accumulated experience in full-aperture

half-meter-scale and tiled meter-scale gratings, as shown in Fig. R-1a.

The engineering project for full-aperture meter-scale grating is in progress. At present, our team has the capability to expose meter-scale gratings. Fig. R-1b shows a 1650 mm \times 1120 mm off-axis mirror in the exposure system⁵. In addition, the meter-scale gratings will be completed in 2023-2024 in SIOM.

Fig. R-1 **a** Meter-scale gold gratings from SIOM. **b** Photograph of a 1650 mm \times 1120 mm off-axis parabolic mirror.

2. This work is in the proof-of-principle stage for the 400 nm ultra-broadband gratings.

Fig. R-2 Fabrication tolerance of **a** line density, **b** duty cycle, **c** shape factor, and **d** depth of the 1443 lines/mm ultra-broadband grating. The yellow rectangle indicates the manufacturing accuracy of the

current meter-scale grating.

Fig. R-2 illustrates the fabrication tolerance of the 1443 lines/mm ultra-broadband grating and the manufacturing accuracy of our team's meter-scale grating. The initial design grating structure is a line density of 1443 lines/mm, duty cycle of 0.634, shape factor of 1.91, and depth of 225 nm. The manufacturing accuracy is indicated by the yellow rectangle, where the accuracy of line density, duty cycle, shape factor, and depth are ± 0.5 lines/mm, ± 0.02 , ± 0.3 , and $\pm 3\%$. Obviously, the ultra-broadband grating is compatible with today's available grating manufacturing. **The above information has been added to Supplementary Note 2.**

As shown in Fig. R-3a, we have fabricated the 1400 lines/mm gold gratings with an aperture of 200 mm \times 150 mm. Fig. R-3b shows that the diffraction efficiency is uniform over the surface.

Fig. R-3. **a** Photograph of a 200 mm \times 150 mm gold grating. The 1400 lines/mm grating is designed to achieve high efficiency from 810 to 1010 nm. **b** Measured diffraction efficiency over the grating surface.

Consequently, future work involves extending our results to meter-scale gratings for other groups or companies worldwide.

References

1. Perry, M. D. et al. Petawatt laser pulses. *Opt. Lett.* **24**, 160-162 (1999).
2. Zeng, X. et al. Multi-petawatt laser facility fully based on optical parametric chirped-pulse amplification. *Opt.*

Lett. **42**, 2014-2017 (2017).

3. Cartlidge, E. The light fantastic. *Science*. **359**, 382-385 (2018).
4. Radier, C. et al. 10 PW peak power femtosecond laser pulses at ELI-NP. *High Power Laser Sci. Eng.* **10**, (2022).
5. <http://stcsm.sh.gov.cn/xwzx/mtjj/20220510/6d61f5ff99ce48c5965dcea6d329cab7.html>.

Question 2. There is no evidence for uniformity of diffraction efficiency.

Answer 2: Thanks for pointing this out.

1. Diffraction efficiency map of small size ultra-broadband gold grating.

In a scanning photometry map, the grating was measured over an area of 46 mm × 46 mm with a laser beam size of 2 mm, a step size of 2 mm. Each diffraction efficiency map contains 576 points, of which the number of single-point sampling is 5. 1443 and 1527 lines/mm ultra-broadband gold gratings were tested at 50° and 62° in TM polarization, respectively.

Fig. R-4 Scanning photometric diffraction efficiency map of 50 mm × 50mm samples at 950 nm.

1443 lines/mm grating: The area with diffraction efficiency higher than 90% accounts for 95.5% of the total measured area, and the average efficiency is 92.7% with a standard deviation of 0.99% (Fig. R-4 left map).

1527 lines/mm grating: The efficiency map is extremely uniform over the central 93% of the grating area, exhibiting an average efficiency of 93.8% with a standard deviation of 0.73% (Fig. R-4 right map).

The relevant data for uniformity of diffraction efficiency has been added in the main article.

2. Diffraction efficiency map of meter-scale gold grating.

A detailed description of the effect of diffraction efficiency map of meter-scale gold gratings on compressed and focused pulses has been added in **Supplementary Note 3**.

The revised version is as follows:

Diffraction efficiency map vs. Compressed and focused pulse. Owing to the limited size of our original exposure system, the tiled meter-size grating was accepted. Fig. S8 shows the measured diffraction efficiency map of a 970 mm × 370 mm grating in our group⁸. The efficiency map is extremely uniform over the central 84% of the grating area, exhibiting an average efficiency of 85.8% with a standard deviation of 0.103%. As stated in Supplementary Note 2, the aperture of ultra-broadband gratings can be scaled to meter size or even larger in engineering. Consequently, to investigate the effect of the diffraction efficiency map of the ultra-broadband grating on the designed near-single-cycle 100 PW laser, Fig. S8 can be directly scaled from 970 mm × 370 mm to 870 mm × 600 mm and 1370 mm × 600 mm, respectively. Moreover, the pulsed beams before and after focusing were re-simulated.

Fig. S8 Scanning photometric diffraction efficiency map of a 970 mm × 370 mm at 750 nm. The blue area displays low diffraction efficiency owing to the seam.

Figure S9 shows the effects of the weak non-uniformity and seam of the diffraction efficiency map (see Figure S8) on the 3D spatiotemporal structure of the compressed and focused pulse beam. The following simulations can be divided into two main categories according to whether or not to consider the existence of seams in ultra-broadband gratings.

Fig. S9 Supported pulsed beams with imperfect gratings. (a, c, e, g) Compressed and (b, d, f, h) focused pulsed beams. The ultra-broadband gratings (a-b) G1-G4 in the compressor were considered to load the diffraction efficiency map without seams in Fig. S8. The ultra-broadband gratings (c-d) G1 and G4, (e-f) G2 and G3, and (g-h) G1-G4 in the compressor were considered to load the diffraction efficiency map with seams in Fig. S8.

If seams are not considered (we have now built an upgraded meter-sized exposure system), Fig. S9 (a) and (b) shows the compressed and focused pulse beams when diffraction efficiency maps are added to the four gratings G1-G4. The simulated 3D spatiotemporal structures have no significant changes compared to Fig.S5 (a) and (b).

Conversely, Fig. S9 (c), (e), and (g) simulate the compressed pulse beams in the presence of a seam in Fig. S8. When the non-uniform diffraction efficiency map is loaded on G1 and G4, a gap in the x-y plane appears in the compressed pulse beam [see Fig. S9(c)]. In addition, the compressed pulsed beam exhibits spatiotemporal modulation in the x-t plane [see Fig. S9 (e)] when G2 and G3 with the non-uniform diffraction efficiency maps. Accordingly, the compressed pulsed beam has a gap in the x-y plane and spatiotemporal modulation in the x-t plane [see Fig. S9 (g)], where G1-G4 all loaded the diffraction efficiency map with seams.

However, owing to the narrow width of the seam, the 3D spatiotemporal structure of the focused pulsed beam [see Fig. S9 (d), (f), and (h)] almost keeps unchanged compared to Fig.S5 (b).

Fig. S10 Degradation of focused peak intensity. The focused peak intensity with perfect gratings (i.e., uniform diffraction efficiency) is normalized to 1. Results of Figs. Figs. S9 (b), (d), (f), and (h) are 0.96, 0.85, 0.87, and 0.75, respectively.

SHANGHAI INSTITUTE OF
OPTICS AND FINE MECHANICS
CHINESE ACADEMY OF SCIENCES
Shanghai 201800 CHINA

P. O. Box 800-211 Shanghai
Tel: 0086-21-69918000
Fax: 0086-21-69918800
<http://www.siom.cas.cn>

Nevertheless, the focused peak intensity is different in the above cases. Here, the focused peak intensity in Fig. S5 (b) (with perfect gratings) is normalized to 1. Accordingly, Fig. S10 shows the focused peak intensities in Figs. S9 (b), (d), (f), and (h) are 0.96, 0.85, 0.87, and 0.75, respectively. The seam in tiled gratings can reduce the focused peak intensity. Typically, large gratings are required at the second and the third gratings G2 and G3 (see Fig. S1), and then the normalized degradation would be around $1 - 0.87 = 0.13$. However, with a sufficiently large exposure system, the normalized degradation induced by the diffraction efficiency map (with no seam) is only around $1 - 0.96 = 0.04$.

References

8. Z. Yu, W. Shenghao, L. Shijie, S. Jianda, J. Yunxia & X. Zhilin. Fast measurement technique for obtaining the diffraction efficiency and its uniformity of a large-aperture pulse compression grating. Proc.SPIE. 2019. p. 1083918.

Question 3. The authors never attempted to compress a broadband laser pulse with this grating and measured its pulse contrast. It is far from obvious that the surface quality of this grating could support compression into such a short pulse without producing a group delay ripple that greatly affects the peak power and could cause excessive spatiotemporal distortion.

Answer 3: Thanks for the comment and query.

1. First, experiments to compress broadband laser pulses by ultra-broadband gratings are in the planning. But I'm sorry. At this stage, it is not possible to test the characteristics of the compressed pulses directly in the experiment because of the lack of an ultra-broadband light source. However, the spectral properties of the grating were experimentally evaluated. For this article, these results are sufficient.
2. Thank you for your concern about the compressed pulse characteristics. **Detailed results have been added in in Supplementary Note 4.**

The revised version is as follows:

Diffraction wavefront vs. Compressed and focused pulse. Fig. S11 shows the measured diffraction wavefront of a 970 mm×370 mm tiled grating. The wavefronts of the left and right parts that make up the tiled grating were measured separately. The measurement wavelength λ is 632.8

nm. The PV values of the left and the right gratings are approximately $\lambda/3$ and $\lambda/2$. In our current manufacturing technology, the wavefront presents a modulation of concentric rings. As stated in Supplementary Note 2, the aperture of ultra-broadband gratings can be scaled to meter size or even larger in engineering. Consequently, to investigate the effect of the diffraction wavefront of the ultra-broadband grating on the designed near-single-cycle 100 PW laser, the 3D spatiotemporal structures of compressed and focused pulsed beams were simulated directly using different concentric ring wavefronts in the following results.

Fig. S11 Measured diffraction wavefront of a 970 mm \times 370 mm tiled grating. The PV values of the left and the right gratings are around $\lambda/3$ and $\lambda/2$. The measurement wavelength λ is 632.8 nm.

Since the wavefronts of G1 and G4 introduce only wavelength-independent distortion^{6,9}, which can be easily eliminated by the deformable mirror, only the wavefronts of G2 and G3 in the compressor are considered in the following analysis.

Figs. S12 (a-f) show the 3D spatiotemporal structure of the compressed and focused pulse beams at a PV value of $\lambda/3$. When the modulation period of the wavefront is 800 mm [see Fig. S12 (a)], both the compressed and the focused pulsed beams generate spatial distortions in the y-t plane and spatiotemporal distortions in the x-t plane [see Fig. S12 (b) and (c)]. In contrast, when the modulation period of the wavefront is 200 mm [see Fig. S12 (d)], distortions become more severe for both the compressed and the focused pulsed beams [see Fig. S12 (e) and (f)] compared to the

results in Fig. S12 (b) and (c).

Fig. S12 Supported pulsed beams with imperfect gratings. (a, d, g, j) Produced diffraction wavefront, (b, e, h, k) compressed and (c, f, i, l) focused pulsed beams. (a, g) The concentric ring wavefront has an 800 mm modulation period and (d, j) 200 mm modulation period. The concentric ring wavefront has (a-f) a $\lambda/3$ PV and (g-l) a $\lambda/2$ PV. The measurement wavelength λ is 632.8nm.

Similarly, Figs. S12 (g-l) display the compressed and focused pulse beams at a PV value of

$\lambda/2$. When the modulation period of the wavefront is 800 mm [see Fig. S12 (g)], distortions [see Fig. S12 (h) and (i)] are slightly deteriorated compared to the results in Fig. S12 (b) and (c). And, compared to the results in Fig. S12 (e) and (f), distortions in Fig. S12 (k) and (l) also deteriorate with a 200 mm modulation period [see Fig. S12 (j)].

Fig. S13 illustrates the values of the normalized focus peak intensity in the above cases. Here, the focus peak intensity in Fig. S5 (with perfect grating) is normalized to 1. The normalized peak focus intensity decreases from 0.71 to 0.60 and from 0.66 to 0.28 as the modulation period decreases from 800 mm to 200 mm at a PV value of $\lambda/3$ and $\lambda/2$, respectively. For the degradation of the normalized focus peak intensity to be less than 50%, the modulation period should be larger than 200 mm, and the modulation PV should be less than $\lambda/2$, which can be satisfied by our current grating manufacturing capability. In addition, this kind of wavefront distortions can now be controlled or pre-compensated^{9,10}, and the method is shown in Fig. S14 and introduced in Refs. [9, 10].

Fig. S13 Degradation of focus peak intensity. The focused peak intensity with perfect gratings (i.e., flat wavefront $L = \infty$, $PV = 0$) is normalized to 1. Results of Figs. S12 (c) ($L = 800$ mm, $PV = 1/3 l$), Figs. S12 (f) ($L = 200$ mm, $PV = 1/3 l$), Figs. S12 (i) ($L = 800$ mm, $PV = 1/2 l$) and Figs. S12 (l) ($L = 200$ mm, $PV = 1/2 l$) are 0.71, 0.60, 0.66, and 0.28, respectively. L , modulation period; $l = 632.8$ nm.

Fig. S14 Pre-compensation of grating wavefronts in the compressor. Reproduced with permission.¹⁰ Copyright 2022, Wiley. Wavelength-dependent wavefront correction is introduced and imaged from a small compressor into the main compressor.

References

1. Li, Z., Liu, J., Xu, Y., Leng, Y. & Li, R. Simulating spatiotemporal dynamics of ultra-intense ultrashort lasers through imperfect grating compressors. *Opt. Express* 30, 41296-41312 (2022).
2. Li, Z. & Kawanaka, J. Complex spatiotemporal coupling distortion pre-compensation with double-compressors for an ultra-intense femtosecond laser. *Opt. Express* 27, 25172-25186 (2019).
3. Li, Z., Leng, Y. & Li, R. Further Development of the Short-Pulse Petawatt Laser: Trends, Technologies, and Bottlenecks. *Laser Photonics Rev.* 17, 2100705 (2022).

Question 4. The authors did not measure the laser damage threshold for their samples, which is critical for the proposed use case.

Answer 4: Thanks for pointing this out.

The Station of Extreme Light (SEL) received the maximum support in the international review meeting on July 14th, 2022. Fortunately, we carried out the laser-induced damage test of 400-nm ultra-broadband gold gratings in the front end for the SEL, which shows the widest spectral width, and the shortest pulse duration among all the 100TW-level OPCPA systems. The photograph below shows the site of the damage experiment.

Fig. Photograph of the damage-test table in the front end for the Station of Extreme Light (SEL).

Detailed damage testing experiments have been added to the main article.

The revised version is as follows:

“Meanwhile, the 1-on-1 damage test (see Method section) was performed to investigate the initial failure state of the ultra-broadband gratings in the front end for the Station of Extreme Light (SEL). The front end provides a laser pulse with the widest spectral width and the shortest pulse duration among all the 100-TW-level OPCPA systems at present. 1443 and 1527 lines/mm ultra-broadband gold gratings exhibit the surface LIDT of 350 mJ/cm^2 and 272 mJ/cm^2 (Fig. 8d), respectively. These values are sufficient to meet the LIDT requirements of current compressors. Future work aims to perform an LIDT test on the meter-scale grating with a more broadband laser in the vacuum.”

“Laser-induced damage test

The LIDT tests of the 400-nm ultra-broadband gold gratings are assessed in SIOM. The layout of the damage test was developed from the front end for the SEL, as shown in Fig. 9a. This front end was an LBO-based OPCPA system, capable of generating a 13.4 fs pulse with a bandwidth over 200 nm near the center wavelength of 925 nm at 0.1 Hz.

Fig.9 Damage-test table in the front end for the Station of Extreme Light (SEL). **a** Layout of the test facility. HR: High reflectivity mirror; HWP: half-wave plate; TFP: thin-film polarizer; L: lens; I: iris; T: translation stage; M: microscope; G: grating; S: site (indicated by two yellow circles). The pulse width, spectrum, and beam profile were measured in S1 or S2. **b** Measured and FTL pulse duration of the compressed laser pulse in S1. **c** Measured spectrum in S1 and S2. **d** Beam profile in S2.

A pair of 200 nm broadband thin-film polarizers has been self-developed with a 90: 1 extinction ratio. An achromatic half-wave plate (Thorlabs AHWP10M-980) in conjunction with two polarizers acted as a fluence-variable-ratio attenuator. The laser polarization was further controlled by another half-wave plate. For the damage test, the beam was focused with a 1500 mm long focal-length lens. The beam profile on the focal plane was imaged onto a charge-coupled device (CCD, Ophir SP907), and the pulse duration was measured by a Wizzler (Fastlite, USP4). The gratings were mounted on a motorized translation stage (LianYi XZM200H-150) at the focus. During the test, the irradiated sites of gratings were observed using a long working distance and

SHANGHAI INSTITUTE OF
OPTICS AND FINE MECHANICS
CHINESE ACADEMY OF SCIENCES
Shanghai 201800 CHINA

P. O. Box 800-211 Shanghai
Tel: 0086-21-69918000
Fax:0086-21-69918800
<http://www.siom.cas.cn>

high magnification zoom microscope objective equipped with a CCD camera (XiangQue MetOPT800). A self-programmed procedure controlled the damage test setup.

As shown in Fig. 9b, the compressed pulse width is 15 ± 1 fs in S1. For the design of high-peak-power damage-test systems, self-phase modulation (SPM) must be considered. The spectrum (Fig. 9c) in S1 and S2 shows a relatively weak self-phase modulation effect. The effective beam size is ~ 0.07 mm² in the beam normal.

According to the ISO-21254 LIDT standard, 1-on-1 damage tests were performed in the air. 1443 lines/mm and 1527 lines/mm gratings were tested at 50° and 62° in TM polarization, respectively. Twelve sites were tested for each energy fluence. Note that the LIDTs were provided on the surface and determined by extrapolating the damage probability curve to zero probability.”

In conclusion, this is an interesting and encouraging result that offers a hint of the possible direction in broadband grating development for CPA, and this reviewer is the opinion that it will be of interest to the audience of a more specialized journal.

Answer: We appreciate your positive comments as well as the questions you have raised. Those comments are all valuable and very helpful for revising our manuscript and improving our work. We are pleased to get your positive feedback and enthusiastic discussion!

Answers to Reviewer 2

Reviewer #1 (Remarks to the Author):

The manuscript “400 nm ultra-broadband gratings for near-single-cycle 100 Petawatt lasers” by Yuxing Han et al. provides a new grating design intended for scaling peak powers beyond current 10-PW-level systems by allowing the compression of much larger bandwidths potentially reaching the 100-PW-level. This detailed design and fabrication methods presented here would prove useful to advancing grating technology and benefiting the laser community. I recommend this manuscript for publication in Nature Communications.

Summary:

SHANGHAI INSTITUTE OF
OPTICS AND FINE MECHANICS
CHINESE ACADEMY OF SCIENCES
Shanghai 201800 CHINA

P. O. Box 800-211 Shanghai
Tel: 0086-21-69918000
Fax: 0086-21-69918800
<http://www.siom.cas.cn>

The authors reference key CPA techniques such as Ti:Sa CPA, BBO OPCPA, DKDP OPCPA, and wide-angle noncollinear OPCPA and some of their limitations. Reducing the pulse duration is identified as one of the more straight-forward ways to significantly boost peak power since current 10 PW lasers are nowhere near a single-cycle pulse duration. Current grating technology can support ~200 nm bandwidths in the near-infrared, but this new design will allow improved broadband diffraction efficiency for reaching single-cycle pulse durations. This article presents a design for 400 nm ultra-broadband gold gratings compatible with large azimuthal angles (ie. out-of-plane compressors) developed for near-single-cycle pulse stretching and compression. The authors identified free parameters that may be adjusted in the grating design and proposed a new design based on an optimized parameter space. In addition to optimization the tolerancing was also considered to ensure that the design was realistic to fabricate with common methods. The grating profile evolution was thoroughly studied to find the most reliable fabrication recipe. A 4 wt.% NaOH developer was found to broaden the base width while a 6 wt.% NaOH developer allowed the sharpening of ridges, both of which contributed to enhancing the efficiency over a broader bandwidth. The grating design was incorporated in a wide-angle NOPCPA system that could produce pulses as short as 6 fs, where the diffraction efficiency of the gratings could ideally compress a bandwidth sufficient for a 4 fs FTL pulse duration. The gratings exhibit a high diffraction efficiency > 90% for >400 nm bandwidth from 750 to 1150 nm, facilitating development of next-generation high-peak-power lasers.

Comments:

- Research is thorough and well-organized with appropriate figures showing the grating design compared to measurements of the fabricated gratings.
- The proposed grating design with optimized base width, groove depth, and shape factor is backed up with quality characterization and experimental implementation in an ultra-broadband OPCPA system.
- The fabrication and characterization methods are described in clear detail, which is very useful to others working to develop improved gratings for ultra-broadband lasers.
- The quality of the manuscript text and figures are high.

We are grateful for your effort reviewing our paper and giving an accurate summary of our work. We

SHANGHAI INSTITUTE OF
OPTICS AND FINE MECHANICS
CHINESE ACADEMY OF SCIENCES
Shanghai 201800 CHINA

P. O. Box 800-211 Shanghai
Tel: 0086-21-69918000
Fax:0086-21-69918800
<http://www.siom.cas.cn>

appreciate your positive comments. Thank you very much for your recognition of our work.

To sum up, we have provided a point-to-point response to the comments made by anonymous reviewers. Please do not hesitate to contact us if there are any questions. Thanks again to the reviewers and editors for your hard work! Best wishes to you!

Sincerely,

Author: Yuxing Han

Corresponding authors: Prof. Yunxia Jin, Prof. Zhaoyang Li and Prof. Jianda Shao

REVIEWERS' COMMENTS

Reviewer #1

Remarks to the authors -

This reviewer provided confidential remarks to the editors recommending publication of the manuscript.

SHANGHAI INSTITUTE OF
OPTICS AND FINE MECHANICS
CHINESE ACADEMY OF SCIENCES
Shanghai 201800 CHINA

P. O. Box 800-211 Shanghai
Tel: 0086-21-69918000
Fax:0086-21-69918800
<http://www.siom.cas.cn>

Answers to Reviewer 1

Reviewer #1

Remarks to the authors -

This reviewer provided confidential remarks to the editors recommending publication of the manuscript.

We are grateful for your effort reviewing our paper and giving an accurate summary of our work. We appreciate your positive comments. Thank you very much for your recognition of our work.

Please do not hesitate to contact us if there are any questions. Thanks again to the reviewers and editors for your hard work! Best wishes to you!

Sincerely,

Author: Yuxing Han

Corresponding authors: Prof. Yunxia Jin, Prof. Zhaoyang Li and Prof. Jianda Shao